

# Mild cognitive impairment in type 2 diabetes is associated with fibrinogen-to-albumin ratios

Xinyu Li[1], Qian Wu[1], Yanqi Kong[1] and Chong Lu[2]

[1] Department of Endocrinology, First Affiliated Hospital of Harbin Medical University, Harbin, Heilongjiang, China
[2] Department of Neurology, Heilongjiang Provincial Hospital, Harbin, Heilongjiang, China

## ABSTRACT

**Background:** Cognitive impairment is the main manifestation of diabetes central neuropathy. Currently, there is no effective dementia treatment; early diagnosis and treatment are particularly crucial. Inflammation index fibrinogen-to-albumin ratio (FAR) has been shown to predict complications of type 2 diabetes (diabetic kidney disease and diabetes-related arteriosclerosis), but its relationship with mild cognitive impairment (MCI) in type 2 diabetes (T2D) is undetermined. In this study, we examined the association between the FAR and mild cognitive impairment in type 2 diabetes.

**Methods:** This is a retrospective and cross-sectional study. From January 2022 to December 2022, we have retrieved 328 inpatient medical records for T2D patients hospitalized at the First Hospital of Harbin Medical University from the hospital's electronic system. Subjects' cognitive function was assessed and grouped by the MoCA scales. Subjects' demographic and various laboratory indicators were collected. Using Spearman's bivariate correlation analysis, the FAR and other clinical variables were analyzed for association strength. A multiple linear regression analysis was conducted to determine the independent relationship between FAR and MoCA scores. Multivariate logistic regression was used to analyze the independent relationship between FAR and MCI. The capacity of the FAR to detect MCI was carried using receiver operating characteristic (ROC) analysis.

**Results:** The included participants' ($n = 328$; 61.9% male) mean age was $52.62 \pm 10.92$ years. MoCA scores and MCI prevalence significantly differed ($p < 0.05$) between the four subgroups of FAR quartiles. The FAR and the MoCA score were significantly negatively correlated in the entire population ($p < 0.05$). Based on the multiple linear regression analysis, lnFAR and lnMoCA are significantly correlated ($\beta = -0.449$, t = $-8.21$, $p < 0.05$, R2 = 0.469). In multivariate logistic regression analysis, FAR and MCI were independently correlated after adjusting for covariates (OR 95% CI 34.70 [13.90–86.66]). Finally, the analysis of receptor working characteristics shows that the optimal FAR cut-off value was 0.08 (sensitivity: 95.81%, specificity: 84.47%) for detecting MCI in type 2 diabetes.

**Conclusion:** In type 2 diabetes, the FAR was positive associations with MCI and negative associations with MoCA score. The high FAR was associated with an increased risk of MCI. FAR maybe a appropriate indicator of MCI risk for type 2 diabetes.

Corresponding authors
Xinyu Li, 13796658016@163.com
Chong Lu, 13796658010@163.com

# INTRODUCTION

Diabetes central neuropathy refers to the structural and functional damage of central nervous system neurons and nerve fibres caused by diabetes-related metabolic disorders. Cognitive impairment is the main manifestation of diabetes central neuropathy, which can be divided into mild cognitive impairment (MCI) and dementia according to the severity of the lesion (*Biessels & Despa, 2018*; *Srikanth et al., 2020*). The onset of cognitive impairment caused by diabetes is hidden and can occur in the early stage of diabetes, and there is a lack of therapeutic drugs that can effectively delay and reverse the disease process (*Dove et al., 2021*). Therefore, early screening and diagnosis of cognitive decline in diabetes is a hot and difficult topic in current research. Aβ reflecting brain amyloidosis and Tau protein reflecting neurodegeneration from the cerebrospinal fluidare are the ideal biomarkers for cognitive impairment, but the cerebrospinal fluid is not easy to obtain. So some biomarker from peripheral blood are urgently needed (*Prestia et al., 2015*; *Jack et al., 2010*). Therefore, it is the great significance for the early diagnosis of MCI which for searching for low-cost and noninvasive biomarkers.

Liver-produced fibrinogen and albumin are useful biomarkers for inflammation and haemodynamic changes (*Chapin & Hajjar, 2015*; *Luyendyk, Schoenecker & Flick, 2019*; *Deveci & Gazi, 2021*). Fibrinogen/albumin ratio (FAR) was more effective in predicting inflammation and tumors combined the two indicators (*Omiya et al., 2021*; *Li et al., 2022*). Moreover, FAR is a stronger independent predictor of diabetic kidney disease than FIB and ALB (*Wang et al., 2021*). No investigation has examined the connection between the FAR and the MCI, although FAR has a role in predicting diabetes complications such as the diabetes-related arteriosclerosis and heart failure (*Wang et al., 2022*; *Huang et al., 2022*). A study of the FAR will assist in managing cognitive impairment throughout the course of T2DM if the FAR is found to be associated with the development of MCI in those with T2DM. As a result, this study examined the association between FAR and mild cognitive impairment in type 2 diabetes.

# MATERIALS AND METHODS

## Subjects

This is a retrospective and cross-sectional study. From January 2022 to December 2022, we have retrieved 328 inpatient medical records for T2D patients hospitalized at the First Hospital of Harbin Medical University from the hospital's electronic system. Based on World Health Organization (WHO) criteria from 1999, T2D was diagnosed. Criteria for the diagnosis of diabetes: FPG ≥126 mg/dL (7.0 mmol/L). Fasting is defiined as no caloric intake for at least 8 h. OR 2-h PG ≥200 mg/dL (11.1 mmol/L) during OGTT (the test should be performed as described by WHO, using a glucose load containing the equivalent of 75 g anhydrous glucose dissolved in water). OR A1C ≥6.5% (48 mmol/mol). (the test should be performed in a laboratory using a method that is NGSP certified and

standardized to the DCCT assay). OR in a patient with classic symptoms of hyperglycemia or hyperglycemic crisis, a random plasma glucose ≥200 mg/dL (11.1 mmol/L). Note: DCCT, Diabetes Control and Complications Trial; FPG, fasting plasma glucose; OGTT, oral glucose tolerance test; NGSP, National Glycohemoglobin Standardization Program; WHO, World Health Organization; 2-h PG, 2-h plasma glucose. In the absence of unequivocal hyperglycemia, diagnosis requires two abnormal test results from the same sample or in two separate test samples (*ElSayed et al., 2023*). The following conditions were excluded from the study: (1) T2DM acute complications such as diabetes ketoacidosis, hypertonic hyperglycaemia; (2) severe hypoglycaemia occurring within a week; (3) Lewy body dementia, frontotemporal dementia, brain trauma, intracranial tumours, history of skull surgery, congenital mental retardation, epilepsy, severe depression and other mental disorders that may seriously affect the assessment of cognitive function; (4) individuals who have a history of drug or alcohol dependence and have taken drugs that affect cognitive function in the past 2 months; (5) known systemic conditions that can lead to dementia, such as primary hypothyroidism, niacin deficiency, hypocalcaemia, neurosyphilis, and HIV infection; (6) current use of drugs to increase the utilization rate of synaptic catecholamine, such as monoamine oxidase inhibitors or tricyclic antidepressants; (7) drugs that reduce the utilization of synaptic catecholamines, such as clonidine, are being used (β 1-adrenergic receptor blockers, *etc.*); (8) severe sleep disorders; (9) hearing or visual impairment, psychiatric disorder.

Written informed consent was obtained from all participants in this study. It was approved by the Ethics Committee of the First Affiliated Hospital of Harbin Medical University.

## Data collection

Population characteristics, smoking, drinking and duration of diabetes were collected and education level. Measure blood pressure, height, and weight. BMI is the ratio of weight (kg) to BMI height ($m^2$). Alanine aminotransferase (ALT), aspartate transaminase (AST), γ-glutamyl transpeptidase (GGT), urea nitrogen (BUN), creatinine (Cr), uric acid (UA), albumin (ALB), glucose (GLU), total cholesterol (TC), triglyceride (TG), low-density lipoprotein cholesterol (LDL-C) and high-density lipoprotein cholesterol (HDL-C) were measured used by Beckmancoulter AU5800 Automatic Biochemistry Analyser.
The fibrinogen (FIB) was measured used by Sysmex CA500 Automatic Blood Coagulation Analyzer. And the glycosylated haemoglobin A1c (HbA1c) was measured used by Sysmex HISCL5000 Automatic High Sensitivity Immune Analyzer. All indicators were measured in all subjects after overnight fasting.

## Cognitive function assessment

MCI is diagnosed using diagnostic criteria established by the National Institute on Aging and the Alzheimer's Association (NIA-AA) (*Albert et al., 2011*). MoCA-BJ is a highly sensitive cognitive screening tool that can detect and distinguish patients with MCI from cognitively normal people (*Yu, Li & Huang, 2012*). It is normal to have a MoCA score of 26

or higher (*Yu, Li & Huang, 2012*; *Nasreddine et al., 2005*). In this study, MCI was defined as a score of less than 26.

## Statistical analysis

The data were analyzed using SPSS statistical software version 25.0 (IBM SPSS Inc., Chicago, IL, USA). It was considered statistically significant when a *p* value <0.05 was used. Means + standard deviations (SDs) are presented for variables with normal distributions. Using Spearman's bivariate correlation analysis, the FAR and other clinical variables were analyzed for association strength. To compare continuous variables between groups, chi-square tests were used. To investigate the associations between FAR and MCI and FAR and MoCA scores, Spearman's bivariate correlation analysis was used for the overall population. A multiple linear regression analysis was conducted to determine the independent relationship between FAR and MoCA scores. Multivariate logistic regression was used to analyze the independent relationship between FAR and MCI. The capacity of the FAR to detect MCI was carried using receiver operating characteristic (ROC) analysis.

# RESULTS

## Basic characteristics

The mean age of the 328 participants was 52.62 ± 10.92 years, and 61.9% of the participants were male. In Table 1, the clinical characteristics of the four subgroups according to the FAR quartiles are presented. With the gradual increase of FAR quartile, the MoCA scores significantly decreased, and the proportion of MCI significantly increased ($p < 0.05$). The proportion of males, duration of diabetes, years of education, systolic blood pressure (SBP), ALT, BUN, Cr, eGFR, UA, ALB, FIB, TC, HbA1c, LDL-C and GLU were all significantly different between the four subgroups ($p < 0.05$). The age, diastolic blood pressure (DBP), smoking and drinking status, BMI, AST, GGT, TG and HDL-C levels were no significant differences between the four subgroups ($p > 0.05$).

## The relationship between the FAR and clinical parameters in type 2 diabetes

The FAR correlated significantly with age, duration of diabetes, BUN, Cr, UA, FIB, HbA1c, TC, TGs, and LDL-C (r = 0.144, 0.238, 0.315, 0.397, 0.171, 0.911, 0.24, 0.28, 0.13 and 0.212, respectively; $p < 0.05$), as shown in Table 2, as well as significant negative associations with years of education, ALT, eGFR, ALB and MoCA scores (r = −0.201, −0.201, −0.463, −0.682 and −0.553, respectively; $p < 0.05$).

## The relationship between the FAR and the MoCA score in type 2 diabetes

As shown in Table 3, the FAR and the MoCA score were significantly negatively associated in the T2D. And it was found that FAR and MoCA scores were significantly negatively correlated in the subgroups, including groups under 60 years old and over 60 years old ($p < 0.05$). A multiple linear regression model showed that the FAR and MoCA score were independent in patients with T2D in Table 4. Among patients with T2D, the lnFAR

**Table 1 Clinical characteristics of the study participants.**

| Variables | Q1 | Q2 | Q3 | Q4 | p value |
|---|---|---|---|---|---|
| FAR | 0.06 (0.05,0.06) | 0.07 (0.07,0.08) | 0.09 (0.09,0.10) | 0.12 (0.11,0.15) | |
| FAR (range) | <0.07 | 0.07−0.08 | 0.08−0.11 | >0.11 | |
| n | 82 | 82 | 82 | 82 | |
| Age (years) | 50.14 ± 11.75 | 51.87 ± 12.02 | 54.47 ± 9.72 | 54.96 ± 10.21 | 0.06 |
| Male (n/%) | 63 (76.83%) | 42 (51.22%) | 47 (57.32%) | 51 (62.2%) | 0.006 |
| Years of edacation (year) | 11.13 ± 2.48 | 10.98 ± 3.01 | 9.38 ± 2.54 | 9.24 ± 2.16 | <0.001 |
| Smoking (n/%) | 36 (43.9%) | 31 (37.8%) | 25 (30.49%) | 33 (40.24%) | 0.331 |
| Drinking (n/%) | 42 (51.22%) | 33 (40.24%) | 31 (37.8%) | 33 (40.24%) | 0.304 |
| Diabetic duration (years) | 7.75 ± 6.56 | 7.54 ± 6.60 | 13.11 ± 8.16 | 13.39 ± 8.74 | <0.001 |
| SBP (mmHg) | 132.38 ± 18.83 | 138.63 ± 21.99 | 141.39 ± 19.81 | 142.3 ± 23.94 | 0.013 |
| DBP (mmHg) | 82.67 ± 10.96 | 82.16 ± 11.86 | 82.85 ± 12.33 | 80.89 ± 11.71 | 0.705 |
| BMI (kg/m$^2$) | 25.32 ± 2.89 | 25.24 ± 3.25 | 25.44 ± 3.45 | 25.58 ± 2.96 | 0.909 |
| ALT (U/L) | 22.40 (16.30,34.50) | 20.45 (14.88,30.05) | 18.75 (13.33,26.65) | 16.80 (11.55,21.40) | <0.001 |
| AST (U/L) | 18.35 (14.35,24.55) | 17.70 (15.23,21.28) | 16.65 (14.25,20.58) | 16.55 (12.98,20.93) | 0.09 |
| GGT (U/L) | 28.15 (19.10,39.88) | 25.70 (18.28,39.88) | 28.60 (19.38,39.20) | 24.25 (18.15,36.55) | 0.495 |
| BUN (mmol/L) | 5.67 ± 1.22 | 5.94 ± 1.49 | 6.25 ± 1.89 | 7.20 ± 2.82 | 0.001 |
| Cr (μmol/L) | 61.05 (51.13,71.6) | 57.8 (50.1,66.95) | 56.95 (48.2,71.43) | 69.25 (52.18,89.58) | <0.001 |
| eGFR (ml/min/1.73m$^2$) | 109.14 ± 13.29 | 107.01 ± 13.26 | 104.74 ± 13.39 | 89.92 ± 28.12 | <0.001 |
| UA (mmol/L) | 331.31 ± 90.30 | 309.11 ± 77.09 | 322.94 ± 90.48 | 357.25 ± 101.37 | 0.007 |
| ALB (g/L) | 43.67 ± 3.47 | 42.14 ± 2.88 | 40.8 ± 3.04 | 36.65 ± 5.13 | <0.001 |
| FIB (g/L) | 2.46 ± 0.28 | 3.09 ± 0.27 | 3.79 ± 0.37 | 4.82 ± 0.90 | <0.001 |
| HbA1c (%) | 8.34 ± 1.64 | 8.59 ± 1.72 | 9.37 ± 1.72 | 9.37 ± 2.15 | <0.001 |
| GLU (mmol/L) | 8.06 ± 2.19 | 8.29 ± 2.65 | 10.40 ± 4.14 | 8.19 ± 3.13 | <0.001 |
| TC (mmol/L) | 4.68 ± 0.87 | 4.79 ± 1.07 | 4.80 ± 1.01 | 5.35 ± 1.54 | 0.001 |
| TG (mmol/L) | 1.74 (1.32,2.55) | 1.68 (1.15,2.67) | 1.63 (1.18,2.59) | 1.75 (1.32,2.52) | 0.603 |
| HDL-C (mmol/L) | 1.06 (0.92,1.21) | 1.05 (0.92,1.35) | 1.12 (0.96,1.3) | 1.1 (0.97,1.3) | 0.253 |
| LDL-C (mmol/L) | 2.92 ± 0.78 | 2.90 ± 0.83 | 3.02 ± 0.75 | 3.37 ± 1.15 | 0.002 |
| MoCA (scores) | 27.07 ± 0.99 | 25.77 ± 2.38 | 22.63 ± 2.17 | 22.16 ± 1.72 | <0.001 |
| MCI (n/%) | 0 (0) | 21 (25.61) | 69 (84.15) | 77 (93.90) | <0.001 |

Notes:
Normally distributed values in the table are given as the mean ± SD, skewed distributed values are given as the median (25% and 75% interquartiles), and categorical variables are given as frequency (percentage).
FAR, fibrinogen/albumin ratio; SBP/DBP, systolic/diastolic blood pressure; BMI, body mass index; ALT, alanine aminotransferase; AST, aspartate transaminase; GGT, γ-glutamyl transpeptidase; BUN, urea nitrogen; Cr, creatinine; eGFR, estimated glomerular filtration rate; UA, uric acid; ALB, albumin; FIB, fibrinogen; HbA1c glycosylated haemoglobin A1c; GLU, glucose; TC, total cholesterol; TG, triglyceride; HDL-C, high-density lipoprotein cholesterol; LDL-C, low-density lipoprotein cholesterol.

showed a negative and significant association with the lnMoCA score in fully adjusted Model 4 (β = −0.449, t = −8.21, p < 0.001, R2 = 0.469).

## The association between the FAR and MCI in type 2 diabetes

Using a multivariable logistic regression analysis, we examine the correlation between FAR and MCI in patients with T2D (Table 5). The model without any adjustments indicated that FAR and MCI were significantly associated (OR 95% CI 37.15 [17.24–80.08]). With all

**Table 2 Relationships between the FAR and clinical parameters in patients with T2D.**

| Variables | r | p value |
|---|---|---|
| Age (years) | 0.144 | 0.009 |
| Years of edacation (years) | −0.201 | <0.001 |
| Diabetic duration (years) | 0.238 | <0.001 |
| SBP (mmHg) | 0.069 | 0.215 |
| DBP (mmHg) | −0.045 | 0.421 |
| BMI (kg/m$^2$) | 0.061 | 0.270 |
| ALT (U/L) | −0.201 | <0.001 |
| AST (U/L) | −0.098 | 0.077 |
| GGT (U/L) | 0.02 | 0.717 |
| BUN (mmol/L) | 0.315 | <0.001 |
| Cr (μmol/L) | 0.397 | <0.001 |
| eGFR (ml/min/1.73m$^2$) | −0.463 | <0.001 |
| UA (mmol/L) | 0.171 | 0.002 |
| ALB (g/L) | −0.682 | <0.001 |
| FIB (g/L) | 0.911 | <0.001 |
| HbA1c (%) | 0.24 | <0.001 |
| GLU (mmol/L) | 0.005 | 0.927 |
| TC (mmol/L) | 0.28 | <0.001 |
| TG (mmol/L) | 0.13 | 0.019 |
| HDL-C (mmol/L) | −0.03 | 0.592 |
| LDL-C (mmol/L) | 0.212 | <0.001 |
| MoCa (scores) | −0.553 | <0.001 |

adjustments, FAR and MCI still exhibited an independent association (OR 95% CI 34.70 [13.90–86.66]).

## Parameters for diagnosing MCI

By using ROC analysis, the FAR cut-off value for detecting MCI in T2D patients was determined. For MCI subjects, the FAR AUC was 0.941 95% CI [0.910–0.964], the ALB AUC was 0.738 95% CI [0.686–0.784] and the FIB AUC was 0.924 95% CI [0.890–0.950]. There was an optimal cut-off point for each marker for the diagnosis of MCI: 0.08 for FAR (sensitivity: 95.81%, specificity: 84.47%), 3.31 for FIB (sensitivity: 89.22%, specificity: 86.34%) and 40.3 for ALB (sensitivity: 61.08%, specificity: 77.02%) (Table 6 and Fig. 1). When screening for MCI, the FAR performed better than the FIB or ALB (Table 7).

## DISCUSSION

With the continuous increase in the ageing population, the incidence rate of diabetes and its related cognitive dysfunction is gradually increasing. DM-related cognitive impairment is a significant central nervous system complication of DM patients indicated by decreased cognitive ability. The risk of MCI and AD in DM patients is 1.44 and 2.14 times, respectively (Davis et al., 2021). In clinical practice, mild cognitive impairment is a

**Table 3 Relationships between the FAR and the MoCA score.**

| Variables | Total ($n$ = 328) | | <60 years ($n$ = 245) | | ≥60 years ($n$ = 83) | |
|---|---|---|---|---|---|---|
| | R | $p$ value | r | $p$ value | r | $p$ value |
| MoCa (scores) | 0.553 | <0.001 | 0.542 | <0.001 | 0.559 | <0.001 |

**Note:**
 $r$ spearman's correlation coefficient.

**Table 4 Multiple linear regression models displaying independent associations of lnFAR with lnMoCA in patients with T2D.**

| Models | B [95% CI] | $\beta$ | $t$ | $p$ | R2 for model |
|---|---|---|---|---|---|
| Model 0 | −0.553 [−0.644 to −0.462] | −0.553 | −11.989 | <0.001 | 0.306 |
| Model 1 | −0.459 [−0.545 to −0.374] | −0.459 | −10.578 | <0.001 | 0.448 |
| Model 2 | −0.233 [−0.331 to −0.136] | −0.233 | −4.704 | <0.001 | 0.607 |
| Model 3 | −0.261 [−0.361 to −0.160] | −0.261 | −5.11 | <0.001 | 0.624 |
| Model 4 | −0.449 [−0.556 to −0.341] | −0.449 | −8.21 | <0.001 | 0.469 |

**Note:**
 Model 0: unadjusted model. Model 1: adjusted for years of education, smoking, drinking, diabetic duration, SBP, DBP, and BMI. Model 2: adjusted for AST, GGT, BUN, Cr, eGFR, and UA. Model 3: adjusted for HbA1c, GLU, TC, TGs, HDL-C, and LDL-C.

transitional state between normal ageing cognitive decline and the early stages of Alzheimer's disease and is considered the best stage to prevent or modify the further development of Alzheimer's disease (*Kivipelto, Mangialasche & Ngandu, 2018*).

A chronic low-grade inflammatory response is an important feature of cognitive impairment in patients with DM. It has been shown in numerous studies that mild systemic inflammation increases the risk of cognitive impairment and affects patient outcomes (*Dove et al., 2021*). Studies have shown that in patients with poor cognitive abilities, the levels of biomarkers for inflammation are significantly higher, indicating that there may be risk factors that can alter cognitive impairment (*Smirnov et al., 2022*). Fibrinogen (FIB) is synthesized in the liver and circulates in the bloodstream. It is a pleiotropic protein, *i.e.*, an important coagulation factor, and plays a crucial role in the systemic inflammatory system. Epidemiological data show that plasma FIB content significantly increases with age, weight, waist-hip ratio, and other factors (*Ceriello, 1997*). FIB is a frequently used biomarker in cognitive and inflammatory research. Some research results show that increases in hypercoagulability markers and inflammatory markers may lead to thrombosis, accelerate atherosclerosis, and eventually lead to or aggravate AD (*Pyun et al., 2020*). Diabetes patients have a higher plasma fibrinogen content than normal people, and fibrinogen is associated with cognitive impairment (*Zhuang et al., 2022*). Albumin is the most abundant protein in human extracellular fluid, is an important biomarker reflecting nutritional status; it participates in many biological activity processes in the body, and has the functions of maintaining plasma colloid osmotic pressure, buffering blood acid-base changes, combining and transporting various endogenous and exogenous substances in the body, and acting as an antioxidant. Albumin is also an important negative inflammatory biomarker and a platelet activation and aggregation

**Table 5 Multivariable logistic regression analysis to identify the association of the FAR with MCI in patients with T2D.**

| Models | B | SE | Wald | *p* | OR [95% CI] |
|---|---|---|---|---|---|
| Model 0 | 3.615 | 0.392 | 85.115 | <0.001 | 37.15 [17.24–80.08] |
| Model 1 | 3.539 | 0.441 | 64.295 | <0.001 | 34.42 [14.49–81.75] |
| Model 2 | 1.995 | 0.744 | 7.185 | 0.007 | 7.35 [1.71–31.61] |
| Model 3 | 2.153 | 0.943 | 5.217 | 0.022 | 8.61 [1.36–54.65] |
| Model 4 | 3.547 | 0.467 | 57.71 | <0.001 | 34.70 [13.90–86.66] |

Note:
Model 0: unadjusted model. Model 1: adjusted for years of education, smoking, drinking, diabetic duration, SBP, DBP, and BMI. Model 2: adjusted for AST, GGT, BUN, Cr, eGFR, and UA. Model 3: adjusted for HbA1c, GLU, TC, TGs, HDL-C, and LDL-C.

**Table 6 ROC analysis for ALB, FIB and the FAR.**

| Variables | AUC | SE | 95% CI | Z | *p* value | Standard | Sensitivity (%) | Specificity (%) | Youden index J |
|---|---|---|---|---|---|---|---|---|---|
| ALB (g/L) | 0.738 | 0.027 | [0.686–0.784] | 8.797 | <0.001 | ≤40.3 | 61.08 | 77.02 | 0.381 |
| FIB (g/L) | 0.924 | 0.016 | [0.890–0.950] | 26.658 | <0.001 | >3.31 | 89.22 | 86.34 | 0.756 |
| FAR | 0.941 | 0.014 | [0.910–0.964] | 31.82 | <0.001 | >0.08 | 95.81 | 84.47 | 0.803 |

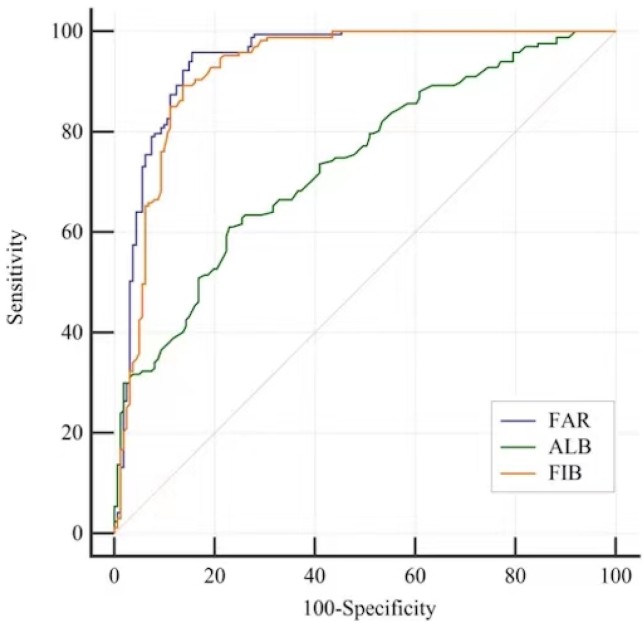

**Figure 1 ROC curves to analyse the ability of ALB, FIB and the FAR to indicate MCI in patients with T2D.** ROC analysis to assess the ability of the FAR to indicate MCI was 0.941 95% CI [0.910–0.964]; Optimal cutoff value of the FAR was 0.08 to indicate MCI; Youden index = 0.803, sensitivty = 95.81% and sepcifcity = 84.47%. The ability of the ALB to indicate MCI was 0.738 95% CI [0.686–0.784]; Optimal cutoff value of the ALB was 40.3 to indicate MCI; Youden index = 0.381, sensitivity: 61.08%, specificity: 77.02%. The ability of the FIB to indicate MCI was 0.924 95% CI [0.890–0.950]; Optimal cutoff value of the ALB was 3.31 to indicate MCI; Youden index = 0.756, sensitivity: 89.22%, specificity: 86.34%.

**Table 7 Comparison of the area under the ROC curve among the FAR, ALB and FIB.**

| Variables | AUC difference [95% CI] | Z | *p* value |
| --- | --- | --- | --- |
| FAR *vs* ALB | 0.204 [0.151–0.256] | 7.631 | <0.001 |
| FAR *vs* FIB | 0.017 [0.005–0.029] | 2.812 | 0.005 |

inhibitor, and it has important physiological functions (*Fanali et al., 2012*). Type 2 diabetes patients had significantly lower serum albumin levels than normal people, and the decrease in albumin level was also closely related to cognitive impairment and dementia (*Kunutsor, Khan & Laukkanen, 2015*). According to clinical research, the dementia patients serum albumin level was significantly lower (*Kim et al., 2020*). Some studies reported that after adjusting for the effects of body mass index, sex, and age, serum albumin levels in AD patients are positively correlated with MMSE scores, and negatively correlated with the onset of AD (*Wu et al., 2020*).

Considering that both plasma FIB and albumin are good biomarkers of inflammation and closely related to cognitive impairment. As an indicator for identifying high-risk groups of diabetes-related MCI, the FAR takes into account inflammation, coagulation, and nutritional status. Numbers studies have confirmed that the FAR is an important marker that reflects systemic inflammatory diseases. Some studies have shown that the FAR can be used as a prognostic indicator for coronary artery severity, stroke, cerebral haemorrhage, tumours (such as pancreatic cancer and liver cancer), COVID-19 and other diseases (*Li et al., 2022*; *Rathore et al., 2022*; *Lin et al., 2022*). As a predictor of diabetic complications like Kidney disease and arteriosclerosis, the FAR was more effective than any single indicator (*Wang et al., 2021*, *2022*). Our research shows that the correlation between FAR and many clinical indicators, such as lipid metabolism, renal function and diabetic duration in Table 2. As important components in the blood, fibrinogen and albumin corresponds closely to type 2 diabetes. Previous studies have confirmed that the level of FIB is positively correlated with the duration of T2D, and with the prolongation of the duration, the risk of FIB increased in elderly T2DM patients (≥60 years old) (*Azad et al., 2014*). The elevated FIB levels were significantly associated with the level of creatinine increased and the glomerular filtration rate (GFR) reduced (*Baggio et al., 2005*). And the total cholesterol, low-density lipoprotein cholesterol, triglyceride may affect the level of fibrinogen through the mechanism such as synthesis rate (*Verschuur et al., 2001*). In type 2 diabetes patients, the level of ALB is positively correlated to renal function. ALB is positively related to eGFR and is an independent influencing factor of eGFR (*Fotheringham et al., 2015*). Therefore, as the fibrinogen-to-albumin ratio, FAR is also closely related to the appeal indicator. We evaluated the relationship between FAR and diabetes-related MCI for the first time and found that with an increase in the FAR, the MoCA score significantly decreased ($p < 0.05$). The proportion of T2D patients with cognitive impairment significantly increased ($p < 0.05$). We also found the FAR and the MoCA score had a significant negative correlation in patients with different ages, and overall T2D patients ($p < 0.05$). Cognitive impairment was significantly associated with

higher FAR values without any adjustment (OR 95% CI 37.15 [17.24–80.08]). After adjusting for all potential covariates that affect cognitive function (such as the diabetic duration, years of edacation, Smoking, Drinking, SBP, DBP, BMI, AST, GGT, BUN, Cr, eGFR, UA, PALB, HbA1c, GLU, TC, TG, HDL-C and LDL-C), the risk of MCI still increased significantly with higher FAR values (OR 95% CI 34.70 [13.90, 86.66]). Based on ROC analysis, there was an optimal cutoff point for the diagnosis of MCI was 0.08 (sensitivity: 95.81%, specificity: 84.47%) for FAR in T2D. This suggests that FAR maybe a appropriate indicator of MCI risk in T2D patients.

Another issue to be discussed is the potential mechanism of association between FAR and MCI in diabetes. Neuroinflammation is an inflammatory reaction in the brain caused by many factors, such as diabetes, energy metabolism disorders, and ageing (*Leng & Edison, 2021*). This inflammation is mainly mediated by cytokines produced by microglia. When microglia are excessively activated, they release a large number of proinflammatory cytokines, thereby damaging neurons, causing synaptic loss and neuronal death and ultimately leading to cognitive impairment (*Dhapola et al., 2021*). Some studies have confirmed that neuroinflammation is the core pathogenesis and one of the main early pathological features of mild cognitive impairment (*Qiu et al., 2021*; *Bradburn, Murgatroyd & Ray, 2019*).

TLR4 is crucial to microglial activation and neuronal damage. In the diabetes cognitive impairment model, the level of TLR4 expression in microglia is increased under a high glucose environment, and the inflammatory response is activated by associated inflammatory factors such as NF-kB and MCP-1 (*Kawamoto et al., 2014*). Excessive activation of inflammatory pathways such as TLR4/NF-κB/NLRP3 and TLR4/Akt/mTOR promotes an increase in the central nervous system inflammatory response and participates in the occurrence and development of cognitive impairment (*Wang et al., 2021*; *Cui et al., 2021*). FIB is a TLR4 ligand that can bind to TLR4 to regulate the aforementioned inflammatory processes, leading to cognitive impairment (*Merlini et al., 2019*). Long-term hyperglycaemia causes the body to be in a long-term chronic inflammatory process. A variety of cytokines produced in this process, such as tumour necrosis factor, interleukin 1 and interleukin 6, may accelerate protein degeneration, increase vascular permeability, and lead to a decrease in serum albumin levels by inhibiting albumin synthesis (*Llewellyn et al., 2010*). Albumin has the characteristic of oxygen free radical capture, as well as the function of multiple binding sites with metal ions, drugs, free fatty acids, and hormones, and has important multiple antioxidant abilities (*Infusino & Panteghini, 2013*). In diabetes, the level of albumin decreases, the production of oxygen free radicals increases, and the oxidative defence ability decreases and the generation of excessive endogenous neurotoxicity, which causes damage to central nervous system function (*Wang et al., 2018*). This study used a MoCA score of <26 as the cut-off point for ROC curve analysis and found that the FAR was better than ALB and FIB for identifying MCI.

There are some limits to this research. 1. In cross-sectional studies, a coincidental relationship between FAR and MCI cannot be demonstrated. 2. It was impossible to account for confounding variables due to a lack of data. 3. In this study, all participants

were Chinese, which may limit the generalizability of the results. In the future, a multicentre prospective study is needed to further verify the ability of the FAR to predict MCI in diabetes.

## CONCLUSIONS

The FAR has a good prediction ability for diabetes MCI, and its potential mechanism may lie in the important role of neuroinflammation in the occurrence and development of mild cognitive impairment.

The FAR can be obtained through coagulation function and liver function teste, which are convenient and economical to detect and provide an effective basis for screening diabetes MCI patients in the future.

### Funding

This study was supported by the National Natural Science Foundation of Heilongjiang Province (No. LH2020H115), the scientific research project of Heilongjiang Health Committee (No. 2019-019, No. 2020-244), the special funds for postdoctoral research of Heilongjiang Province (No. LBH-Q20114), and the Harbin Medical University Affiliated First Hospital Outstanding Young Medical Talent Training Funding Project (No. 2021J14). The funders had no role in study design, data collection and analysis, decision to publish, or preparation of the manuscript.

### Grant Disclosures

The following grant information was disclosed by the authors:
National Natural Science Foundation of Heilongjiang Province: LH2020H115.
Heilongjiang Health Committee: 2019-019, 2020-244.
Heilongjiang Province: LBH-Q20114.
Harbin Medical University Affiliated First Hospital Outstanding Young Medical Talent Training Funding Project: 2021J14.

### Competing Interests

The authors declare that they have no competing interests.

### Author Contributions

- Xinyu Li conceived and designed the experiments, performed the experiments, analyzed the data, prepared figures and/or tables, authored or reviewed drafts of the article, and approved the final draft.
- Qian Wu performed the experiments, prepared figures and/or tables, and approved the final draft.
- Yanqi Kong performed the experiments, prepared figures and/or tables, and approved the final draft.
- Chong Lu conceived and designed the experiments, analyzed the data, authored or reviewed drafts of the article, and approved the final draft.

## Human Ethics

The following information was supplied relating to ethical approvals (*i.e.*, approving body and any reference numbers):

The First Affiliated Hospital of Harbin Medical University granted Ethical approval to carry out the study within its facilities.

## Data Availability

The raw measurements are available in the Supplemental File.

## Supplemental Information

Supplemental information for this article can be found online at http://dx.doi.org/10.7717/peerj.15826#supplemental-information.

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
