# Peer review of "Mild cognitive impairment in type 2 diabetes is associated with fibrinogen-to-albumin ratios"

_PeerJ, doi:10.7717/peerj.15826_

## Round 0.1 · original submission · Minor Revisions

A point to point response letter. Language is needed to improve.

·

Basic reporting

In this retrospective study of 328 type 2 diabetes (T2D) inpatients, we found that a higher fibrinogen-to-albumin ratio (FAR) was associated with an increased risk of mild cognitive impairment (MCI) and lower Montreal Cognitive Assessment (MoCA) scores. The FAR showed independent correlation with MCI even after adjusting for covariates. The optimal FAR cutoff for detecting MCI in T2D was 0.08. These findings suggest that FAR may serve as a valuable indicator for identifying individuals with T2D at risk for MCI.

Experimental design

Limitations:
1. The study population is limited to inpatients from a single hospital, which may not be representative of the broader population of individuals with T2D. A more diverse sample would strengthen the generalizability of the findings.
2. According to the author's description in the article, age is an important factor affecting the occurrence of MCI in patients. It is recommended to remove gender grouping from Table 3 and directly analyze subgroups based on the age.

Validity of the findings

1. The BMI in Table 1 is as high as 40 or above in Q1 and Q2. It is recommended that the authors check the raw data.
2. The problem with both Table 4 and Table 5 statistics is that clinical indicators related to FAR were identified based on Table 2. To demonstrate whether FAR is independently related to MCI or McCa, all relevant indicators should be included simultaneously.
3. In Table 7, it should not be AUC, but AUC difference value.

Reviewer 2 ·

Basic reporting

This study provides valuable insights into the association between FAR and MCI in type 2 diabetes. However, future prospective studies involving diverse populations and incorporating longitudinal assessments are necessary to confirm the findings and establish a causal relationship between FAR and MCI in type 2 diabetes.

Experimental design

The study utilized a relatively large sample size, including 328 inpatient medical records, which increases the robustness of the findings.
The cognitive function assessment was performed using the MoCA scales, a widely recognized and validated tool for evaluating cognitive impairment.
Multiple linear regression analysis and multivariate logistic regression analysis were employed to assess the correlation between FAR, MoCA scores, and MCI while adjusting for relevant covariates.

Validity of the findings

1.In the abstact section the method should be fully described.
2.The introduction logic is poor and need modified .
3.For all parameters, especially the two biomarkers of interest, albumin and fibrinogen, there is no mention of their measurement methods or instruments.
4.The method needs more detail. Please list the T2D diagnostic criteria.
5.Please add the subject selection algorithm as figure 1.
6.The correlation between FAR and many clinical indicators, such as lipid metabolism, renal function and diabetic duration, is found in Table 2. It is suggested that the author explain in the discussion.

Additional comments

This article studied the the association between the FAR and mild cognitive impairment in T2D. Why choosed MoCA-BJ instead of MMSE for the evaluation of cognitive impairment?

·

Basic reporting

The language editing should be made by a professional editing service

Experimental design

The study design should be celarly explained

Validity of the findings

well

Additional comments

Dear Authors,
I reviewed the manuscript entitled "Mild cognitive impairment in type 2 diabetes is associated
with ûbrinogen-to-albumin ratios" meticulousy. Overall idea is interesting. However some points should be clarfied before publication as follows:
- The ethical approval number should be given
- The nature of study should be comprehensively given in methodology. Such as retrospective, prospective, controlled, cross-sectional etc.
- The mechanism of action or possible relation should be more comprhensively discussed. For instance lower albumin level and metabolic effects: Higher fibrinogen levels and thrombotic processes, circulatory effects can be adressed in accordance to previous literature. "- The relationship between fibrinogen to albumin ratio and severity of coronary artery disease in patients with STEMI. Am J Emerg Med. 2016 Jun;34(6):1037-42. doi: 10.1016/j.ajem.2016.03.003.
- Simple blood tests as predictive markers of disease severity and clinical condition in patients with venous insufficiency. Blood Coagul Fibrinolysis. 2016 Sep;27(6):684-90. doi: 10.1097/MBC.0000000000000478"
- Limitations of study should be explained clearly.

---

## Round 0.2 · Minor Revisions

There is a confusing situation about ethical issues, the reviewer said the number is missing. I noticed that the ethical number is lacking because, in some Chinese hospitals, the number of clinical studies is low thereby the ethical committee only provides a dated approval. Please provide more certification for ethical approval, overall, the revising process is well and the revised manuscript meets the publication standard.

·

Basic reporting

Dear Authors,
I re-reviewed the manuscript entitled "Mild cognitive impairment in type 2 diabetes is associate with fibrinogen-to-albumin ratios" meticulously. The revision seems partially made. However, the important points were neglected as below:
The author identified as “It was approved by the Ethics Committee of the First Affiliated Hospital of Harbin Medical University” in the main text. Controversially they responded to the reviewer as below:
This is a research project, so there is no ethical approval number.
It is unacceptable. If the study is approved by a committee then the number should be provided. If the opposite then there is an ethical problem.
Moreover, the language editing and other points are not satisfactory.

Experimental design

.

Validity of the findings

.

---

## Round 0.3 · accepted · Accept

Authors have uploaded the ethical approval with the approval number.